# Modeling high dimensional point clouds with the spherical cluster model

## Abstract

A parametric cluster model is a statistical model providing geometric insights onto the points defining a cluster. The *spherical cluster model* (SC) approximates a finite point set $P \subset \mathbb{R}^d$ by a sphere $S(c, r)$ as follows. Taking $r$ as a fraction $\eta \in (0, 1)$ (hyper-parameter) of the std deviation of distances between the center $c$ and the data points, the cost of the SC model is the sum over all data points lying outside the sphere $S$ of their power distance with respect to $S$. The center $c$ of the SC model is the point minimizing this cost. Note that $\eta = 0$ yields the celebrated center of mass used in KMeans clustering.

We show that fitting a spherical cluster leads to a strictly convex but non-smooth combinatorial optimization problem, and we develop an exact solver based on the Clarke gradient of non-smooth functionals over a suitable stratified cell complex induced by an arrangement of hyperspheres. To the best of our knowledge, our method is the first practical application of the theory of semiflows of convex maps, which generalizes the gradient flows of smooth maps. We present experiments on a variety of datasets ranging in dimension from $d = 9$ to $d = 10,000$, with two main observations. First, our exact algorithm is orders of magnitude faster than BFGS based heuristics for datasets of small/intermediate dimension and small values of $\eta$, and for high dimensional datasets (say $d > 100$) whatever the value of $\eta$. Second, the center of the SC model behave as a parameterized high-dimensional median.

The SC model is of direct interest for high dimensional multivariate data analysis, and holds promises for the design of mixtures.

**Keywords:** subspace clustering, spherical clusters, centerpoints, medians, non smooth optimization.

# 1    Introduction

Our work on the *spherical cluster model* lies at the confluence of three topics: clustering algorithms, parametric cluster models, and high dimensional data analysis.

**Clustering methods.**    Clustering, namely the task which consists in grouping data items into dissimilar groups of similar elements, is a fundamental problem in data analysis at large Xu & Wunsch (2005). Existing clustering methods may be ascribed to four main tiers. *Hierarchical clustering* methods typically build a dendogram whose leaves are the individual items, the grouping aggregating similar clusters Duda & Hart (1973). In *density based clustering* methods, a density estimate is computed from the data, with clusters associated to the catchment basins of local maxima Cheng (1995). Topological persistence may be used to select the significant maxima Chazal et al. (2013). In *spectral clustering* methods, clusters are defined from the top singular vectors of the matrix representing the data (or their similarity) Von Luxburg (2007). *K-means and variants* aim at grouping the data points into a predefined set of $k$ clusters so as to minimize the sum of intracluster variance. Such methods aim at solving a NP-hard optimization problem, and the so-called smart-seeding strategy `k-means++` provides guarantees (in terms of expectation) on the `k-means` functional Arthur & Vassilvitskii (2007). In practice, this strategy is superseeded by a greedy *inertia* based criterion which consists of picking a seed amidst a set of candidates–see Arthur & Vassilvitskii (2007) and the `scikit-learn` implementation of `k-means++`. These methods are related to the problem of fitting (Gaussian) mixtures using Expectation-Maximization Dempster et al. (1977); Kasarapu & Allison (2015). We note in passing that the variety of clustering methods prompted the development of methods to estimate the relevant number of clusters–*e.g.* the *elbow* method Ng (2012), as well as methods to compare two clusterings Cazals et al. (2019).

**Cluster models.**    A central goal of clustering is to provide insights into the geometry of the data. This goal prompted the development of `k-subspaces` clustering techniques, which belong to two tiers. The first one consists of methods in the lineage of (affine) sparse subspace clustering (SSC/ASSC) Elhamifar & Vidal (2013); Soltanolkotabi & Candes (2012); Li et al. (2018). These two step methods write each data point as a sparse linear combination of other data points, and the coefficients found are used to obtain the clusters via spectral clustering. Their correctness hinges on the ability of spectral clustering to separate the clusters, which relies on conditions (*e.g.* the absence of intersection between the affine supports of the clusters) that may not be met in practice. The second tier involves clustering methods using an explicit *i.e.* parametric cluster model Parsons et al. (2004); Wang et al. (2009). These techniques face two difficulties. The first is to avoid overfitting using a complexity penalty (AIC, BIC, MDL, MML) Grünwald (2007), as a richer model always decreases the fitting error–*e.g.* a plane better fits noisy data distributed along a line than the line itself. The second is to obtain the cluster mixture representing the data, a task usually addressed using an Expectation-Maximization procedure  Dempster et al. (1977); Wu (1983). However, the main difficulty for heterogeneous mixtures (*e.g.* clusters of varying dimension) is to navigate in the space of models, a difficult question typically undertaken via (split, merge, delete) operations on the mixture components Kasarapu & Allison (2015), or using a combination of EM and model selection Figueiredo & Jain (2002).

**Geometric centerpoints in data analysis.**    Cluster models providing insights on the geometry of a point set also call for a discussion of high dimensional centerpoints and medians. The classical center of mass of a point set, which minimizes the sum of square distances to data points and is used in `k-means`, admits several important alternatives. The Fermat-Weber point is the point from $\mathbb{R}^d$ minimizing the sum of Euclidean distances to all data points. Unfortunately, this point is hard to compute and unstable Kupitz & Martini (1997); Weiszfeld (1937); Bajaj (1988); Cohen et al. (2016). Building on Helly's theorem, a median can be defined as any point whose Tukey depth is at least $\geq n/(d+1)$ Tukey (1975). (The Tukey depth or halfspace depth of a point $x$ is the smallest fraction of points of any closed half-space containing $x$ Tukey (1975).) It is, however, challenging to compute. The classical randomized algorithm Clarkson et al. (1993) has been derandomized in Miller & Sheehy (2009). The complexity is subexponential in $d$, but to the best of our knowledge, the algorithm is not practical. The projection median is defined by projecting the dataset onto random lines, computing the univariate median for each projection, and computing a weighted average of the data points responsible for these univariate medians Durocher & Kirkpatrick (2009); Basu et al. (2012);

Durocher et al. (2017). It is an elegant, stable and remarkably effective generalization of the univariate median. The projection median also underlies the construction of the Donoho-Stahel estimator for outlier detection Stahel (1981); Donoho (1982); Donoho & Gasko (1992).

**Contributions.** Two types of parametric cluster models have been proposed recently Wang et al. (2009): affine and spherical clusters (SC). The former accommodates potentially unbounded (large) clusters of arbitrary dimension. The latter defines compact (spherical) clusters based on the power distance of points with respect to a sphere whose radius is (a fraction of) the variance of distances to the cluster center–to be determined. However, the uniqueness of the SC center is not established, and no algorithm is presented to compute it. (The calculation presented assumes the center is known, and it solely observes that the result obtained is consistent with the usual center of mass when the fraction of the variance tends to zero Wang et al. (2009).)

Our work, which focuses on the spherical cluster model, is rooted in statistical analysis: identifying an object capturing a global description of the point set. We make three contributions.

First, we establish the SC cluster model is well posed – that is the solution is unique. Second, we present an exact solver using the Clarke gradient on a suitable stratified cell complex defined from an arrangement of hyper-spheres. To the best of our knowledge, our method is the first practical application of the general theory of *semiflows* of convex maps, and may be of independent interest beyond the spherical clustering problem. In our setting, it is numerically tractable and outperforms traditional convex optimization frameworks, particularly in high dimensions, as demonstrated by our experiments. Third, we present experiments showing that the center of the SC model behave as a parameterized high-dimensional median.

These contributions have two direct practical applications and implications. First, the center of SC can be computed efficiently, which is of interest to compute high dimensional centers and/or identify inliers/outliers in high dimensional data analysis. Second, our algorithm and its implementation provide the missing machinery to compute mixtures of spherical clusters in affine subspaces of positive codimension.

All proofs and detailed algorithms are provided in the Supporting Information.

## 2 Parametric cluster models: affine and spherical clusters

### 2.1 Notations and terminology

**Geometry.** Let $D$ be a set of $n$ points in $\mathbb{R}^d$. We consider a partition of $D$ into $k$ clusters $C_1, \ldots, C_k$, with $D_\ell$ the set of points associated to cluster $C_\ell$. The unbiased sample variance for cluster $D_\ell$ of center $c_\ell$ satisfies

$$\hat{\sigma}^2(c_\ell) = \frac{1}{n-1} \sum_{x_i \in D_\ell} \|x_i - c_\ell\|^2 \tag{1}$$

Let $A = c + V$ be an affine space, with $c$ a point in $\mathbb{R}^d$ (think cluster center), and $V$ a vector space. For any point $x \in \mathbb{R}^d$, we denote by $(x - c)_{\|V}$ the orthogonal projection of the vector $(x - c)$ onto $V$, and by $(x - c)_{\perp V}$ the orthogonal projection on $V^\perp$.

When fitting a model, the sum of squared distances from samples to the model is called the *residual sum of squares (RSS)*, or dispersion for short.

Finally, a ball and a sphere of center $c$ and radius $R$ are respectively defined by $\|x - c\|^2 \leq 1$ and $\|x - c\|^2 = 1$.

**Parametric cluster models.** Take $C_\ell$ for $\ell \in [\![1, k]\!]$ and suppose $D_\ell$ is known. Cluster $C_\ell$ is described by the parameter set $\theta_\ell = (\theta_{\ell,1}, \ldots, \theta_{\ell,r})$ and a function $d_\ell : (x, C_\ell(\theta_\ell)) \mapsto d_\ell(x, C_\ell(\theta_\ell))$, that is some distance from a point to the cluster. We call the description of $C_\ell$ by the function $d_\ell$ a *parametric cluster model*. We decompose the clustering problem into two sub-problems concerned with the minimization of a dispersion term based on squared distances:

**Problem. 1** (Cluster optimization)**.** *Let $C_\ell$ be a parametric cluster.* Cluster optimization *is the optimization problem seeking the cluster parameters minimizing the dispersion*

$$\min_{\theta_\ell} \Phi_l, \ with \ \Phi_l = \sum_{x \in D_\ell} d_\ell(x, C_\ell(\theta_\ell))^2. \tag{2}$$

### 2.2 Affine and spherical clusters

**Affine clusters.** As a first generalization of `k-means`, one can consider the distance from a data point to an affine subspace, yielding `k-subspaces` clustering Wang et al. (2009):

**Definition. 1** (Subspace cluster)**.** *Let $A = c + V$ be some affine subspace of $\mathbb{R}^d$ where $c \in \mathbb{R}^d$ is a point and $V$ is an $m$-dimensional linear subspace. The* subspace cluster $C_\ell(A)$ *is a cluster, where the distance from a point $x$ to the cluster is the distance to the subspace :*

$$d(x, C_\ell(A))^2 := d(x, A)^2 = \|(x - c)_{\perp V}\|^2. \tag{3}$$

**Spherical clusters.** As noticed in Introduction, affine clusters may be confounded by noise, and suffer from their non compact nature. This latter aspect can be taken care of using *spherical clusters*. To see how, recall that the power of a point $x$ with respect to a sphere $S(c, r)$ is defined by $\pi(x, S) = \|x - c\|^2 - r^2$. Following Wang et al. (2009), we define:

**Definition. 2** (Spherical cluster)**.** *Let $\eta \in (0, 1)$ be a hyperparameter, and let $c_\ell$ be a point called the* cluster center*. Given the set $D_\ell$, the distance function associated to the* spherical cluster $C_\ell(c_\ell)$ *reads as*

$$d(x, C_\ell(c_\ell))^2 := \max\left(0, \|x - c_\ell\|^2 - \eta\hat{\sigma}^2(c_\ell)\right) = \max\left(0, \pi(x, S(c_\ell, \sqrt{\eta}\hat{\sigma}(c_\ell)))\right). \tag{4}$$

The rationale of this definition is that one wishes to find the center minimizing the cost of outliers–points outside the spherical cluster.

### 2.3 Spherical clusters: discussion

Cluster optimization (Pb. 1) for the spherical cluster model requires optimizing the following functional:

$$F_\eta(c) := \sum_{x_i \in D_\ell} \max\left(0, \|x_i - c\|^2 - \frac{\eta}{n-1} \sum_{x_j \in D_\ell} \|x_j - c\|^2\right) \tag{5}$$

$$= \sum_{x_i \in D_\ell} \max\left(0, \pi(x_i, S_c)\right). \tag{6}$$

Note that the power distance $\pi(x_i, S_c(c, R_c))$ is taken with respect to the sphere centered at $c$ and with squared radius $R_c^2 = \eta/n{-}1 \sum_{x_j \in D_\ell} \|x_j - c\|^2$.

A central result of our work is to show that the optimal center $\text{opt}_{\text{Exact}}$ is unique and can be computed efficiently.

This optimization problem calls for several important comments.

**Using the (quadratic) power distance.** Using the power distance rather than the Euclidean distance serves two purposes. First, when $\eta \to 0$, the optimal center $\text{opt}_{\text{Exact}}$ converges to the usual center of mass of the point set, namely the point minimizing the sum of squared distances. Thus, the centerpoint $\text{opt}_{\text{Exact}}$ may be seen as a parameterized center of mass. Second, using the squared distance simplifies the algebraic calculations carried out in the next sections, and alleviates constraints on number types to obtain a robust implementation. This design choice is actually common. On the one hand, we have recalled in Introduction the difficulty of computing the Fermat-Weber point instead of the center of mass (COM). One the other hand, one should recall that the most general affine Voronoi diagrams are power diagrams (replacing the Euclidean distance by the power distance), while Voronoi diagrams using a multiplicative version of the Euclidean distance are much more complex to handle Boissonnat et al. (2006).

**Radius $R_c$ and its dependency to the centerpoint.** The radius used to define the sphere is not fixed but depends on the location of the centerpoint $c$. It is this interplay which makes the problem difficult. Strictly speaking, it is the *parameterized* std deviation of distances from $c$ to the data points. We will abuse terminology and plainly speak of the *distance variance/std deviation* to the center $c$.

The dependency of $R_c$ to $c$ introduces a subtle mix between inliers and outliers: inliers lie inside $S_c$ and incur zero cost, whereas outliers lie outside and pay the power distance to $S_c$. The optimization problem is therefore ruled by the balance between these two point sets.

**Non-smooth convex optimization problems.** Convex optimization has been extensively studied over the past decades. Our problem involves the optimization of a non-smooth convex function, for which standard methods such as gradient descent may perform poorly in certain cases. Theoretical guarantees are also weaker: classical bounds on the number of iterations required to reach a point within distance $\varepsilon$ of a minimizer grow much faster as $\varepsilon \to 0$ than in the smooth setting Bubeck (2015). As we shall see, our constructive proof and the associated algorithm avoid this caveat.

**Hyper-parameter $\eta$ and balance between inliers vs outliers.** The parameter $\eta$ determines the radius $R_c$ and therefore the functional to be optimized. Varying $\eta \in (0, 1)$ yields a one parameter family of optimization problems. Let #outliers(c) be the number of outliers with respect to a sphere of radius $R_c$ centered at $c$. Upon varying $\eta$, two statistics of interest to assess the role of $\eta$ are the (i) the *average outlier cost* $F_\eta(\text{opt}_{\text{Exact}})/\#\text{outliers(SC)}$, and (ii) the *outlier ratio* $\#\text{outliers(COM)}/\#\text{outliers(SC)}$ which compares the number of outliers yielded by our model and that associated with the usual COM. See Section 5.2 for details.

**Spherical clusters and their merits.** The fundamental motivation underlying the spherical cluster model is rooted in statistical analysis: identifying an object capturing a global description of the point set. In a broad machine learning / data analysis context, this model is attractive for several reasons.

First, its cluster center defines a high dimensional centerpoint which can be compared to the usual center of mass and high dimensional medians. Second, this cluster model provides a natural way to identify inliers and outliers, and the *scale* at which they appear when varying $\eta$. Third, the existence of an efficient (exact) algorithm to compute it paves the way to mixture design algorithms–to be explored in further work.

## 3 Spherical cluster optimization

We study problem of Eq. 5, simply denoting $F_\eta$ as $F$ since $\eta$ is fixed.

### 3.1 Functional decomposition and geometry of the sub-functions

For a fixed data set $D_\ell$, we aim at minimizing over $\mathbb{R}^d$ the map $F_\eta(c)$ from Eq. 5.

To study the previous function, for each $x_i \in D_\ell$, let

$$f_{\eta,x_i}(c) := \|x_i - c\|^2 - \eta\frac{1}{n-1}\sum_{x_j \in D_\ell} \|x_j - c\|^2. \tag{7}$$

so that

$$F_\eta(c) = \sum_{x_i \in D_\ell} \max(0, f_{\eta,x_i}(c)). \tag{8}$$

We first analyze the sub-functions and $f_{\eta,x_i}$ in order to analyze the main function $F_\eta$. In the sequel, we assume that (i) the set $D_\ell$ is fixed, (ii) $x_i \in D_\ell$, (iii) $0 < \eta < 1 - 1/n$ (iv) $\eta$ is fixed, so that we drop $\eta$ from the notations (e.g writing $F, f_{x_i}$ instead of $F_\eta, f_{\eta,x_i}$ to ease notations).

Studying the function $f_{x_i}$ benefits from the geometry of the following *sink region* yielding a null cost:

**Definition. 3.** *The* sink region $B_{x_i}$ *is the set over which $f_{x_i}$ does not contribute to $F$, that is $B_{x_i} := f_{x_i}^{-1}(-\infty, 0]$. We denote $S_{x_i}$ its topological boundary.*

Remark that since $\eta < 1 - \frac{1}{n}$ the intersection $B$ of all $B_{x_i}$ is necessarily empty, as any $x$ belonging to this set would verify $\|x - x_i\|^2$ strictly lower than the average $\frac{1}{n} \sum_{x_i \in D_\ell} \|x - x_i\|^2$.

The following results from an elementary calculation.

**Lemma. 1** (Geometry of $B_{x_i}$). *Let $\eta' = \frac{n-1}{n}\eta$. Each map $f_{x_i}$ is proportional to a spherical power, and takes the form*

$$f_{x_i}(c) = (1 - \eta')\left(\|c - c_i\|^2 - R_i^2\right) \tag{9}$$

*Putting $\bar{x} := \frac{1}{n}\sum_{x_j \in D_\ell} x_j$, the center $c_i$ and the radius $R_i$ of said sphere satisfy the following.*

$$\begin{cases} c_i = \dfrac{x_i - \eta'\bar{x}}{1 - \eta'}, \\ R_i^2 = \left\|\dfrac{x_i - \eta'\bar{x}}{1 - \eta'}\right\|^2 - \dfrac{\|x_i\|^2 - \frac{\eta'}{n}\sum_{x_j \in D_\ell}\|x_j\|^2}{1 - \eta'}. \end{cases} \tag{10}$$

*As a consequence the sink region $B_{x_i}$ is a non-empty closed ball of $\mathbb{R}^d$, and $S_{x_i}$ is its associated sphere.*

As an immediate corollary, $\max(0, f_{x_i})$ is a convex map. In $\mathbb{R}^d \setminus B_{x_i}$, it is quadratic with gradient $\nabla f_{x_i}(c) = 2(1 - \eta')(c - c_i)$, while being identically zero inside $B_{x_i}$.

### 3.2 Arrangement of hyper-spheres underlying the objective function

From the previous lemma, $F$ is also continuous, convex and piecewise quadratic. Finding the optimal cluster center requires understanding of the relationship between all sink regions.

**Arrangement.** An *arrangement* of hyper-surfaces is a decomposition of $\mathbb{R}^d$ into equivalence classes of points using their position with respect to these hyper-surfaces Halperin & Sharir (2017). We apply this concept to the spheres bounding the sink regions (Lemma 1).

For $x \in \mathbb{R}^d$ and $i \in [\![1, n]\!]$, consider the following signature which states whether point $x$ lies outside/on/inside the spheres $S_{x_i}$. It is a length $n$ vector with one entry in $\{-1, 0, 1\}$ for each sink-defining ball:

$$\sigma(x) := (sign(f_{x_1}(x)), \ldots, sign(f_{x_n}(x))). \tag{11}$$

The signature defines an equivalence relation, where two points are equivalent if they have the same signature. We call *cells* the equivalence classes, and we use the notation $\mathcal{C}$ to denote them. By definition, cells are non-empty and characterized by the three-set partition $I^+(\mathcal{C}), I^0(\mathcal{C}), I^-(\mathcal{C})$ of $[\![1, n]\!]$, where the sets are defined respectively as the sets of indices $i$ where $f_{x_i}$ are positive, zero, and negative. Note that generically, $\tau + 1 \le d$ spheres in dimension $d$ intersect along an $l = d - (\tau + 1)$ sphere; thus we let the *dimension* of a cell $\mathcal{C}$ be the number $d - \#I^0(\mathcal{C})$. Cells of dimension $d$ are said to be *fully dimensional* and are open subsets of $\mathbb{R}^d$, while others are said to be of *positive codimension*.

**Combinatorial decomposition of $F$.** On a cell $\mathcal{C}$, $F$ is determined by the value of $f_{x_i}$ where $i$ ranges among $I^+(\mathcal{C})$. More precisely, we have

$$F_{|C}(c) = \sum_{i \in I^+(\mathcal{C})} f_{x_i}(c) = f_{I^+(\mathcal{C})}(c), \tag{12}$$

where $f_J := \sum_{i \in J} f_{x_i}$ for any subset $J$ of $[\![1, n]\!]$. In the same vein, we put $S_J := \bigcap_{i \in J} S_{x_i}$ so that in a generic configuration of spheres any cell with non empty $I^0$ is a relatively open subset of $S_{I^0}$. We define $c_J$ to be the center of mass of all $c_i$ where $i$ ranges among $J$.

$$c_{I^+(J)} := \frac{1}{\#J} \sum_{i \in J} c_i. \tag{13}$$

Putting $R_J^2 := \|c_J\|^2 + (\#J)^{-1}\left(\sum_{i\in J} R_i^2 - \|c_i\|^2\right)$, straightforward computations yield

$$f_J(c) = (1-\eta')\#J\left[\|c - c_J\|^2 - R_J^2\right]. \tag{14}$$

Since any full-dimensional cell $\mathcal{C}$ is open, $F$ is twice differentiable in $\mathcal{C}$ with gradient and Hessian as follows:

$$\begin{cases} \nabla F_{|\mathcal{C}}(c) = 2(1-\eta')\#I^+(\mathcal{C})\left(c - c_{I^+(\mathcal{C})}\right) \\ HF_{|\mathcal{C}}(c) = 2(1-\eta')\#I^+(\mathcal{C})\mathrm{Id}. \end{cases} \tag{15}$$

### 3.3 Strict convexity and optimization

We have seen that $B = \bigcap_{x_i \in D_\ell} B_{x_i}$ is empty, and from previous computations the Hessian of $F$ is almost-everywhere positive definite outside of $B$. This leads to the strict-convexity (and thus, well-posedness) of the problem.

**Theorem. 1** (Strict convexity of $F$). *Let $D_\ell$ be a set of $n$ points of $\mathbb{R}^d$, with at least two distinct points, and assume that $0 < \eta < 1 - \frac{1}{n}$. Then the associated $F$ map is $2(1-\eta')$-strongly convex on $\mathbb{R}^d$, and is a fortiori strictly convex. Its minimization problem admits exactly one solution in $\mathbb{R}^d$.*

**Minimum of $F$ on a full-dimensional cell.** From Eq. (15), the minimum of $F$ is attained on a full-dimensional cell $\mathcal{C}$ if and only if the gradient of $F$ vanishes in $\mathcal{C}$, leading to the following characterization.

$$\operatorname*{argmin}_{\mathbb{R}^d} F \in \mathcal{C} \iff c_{I^+(\mathcal{C})} \in \mathcal{C}. \tag{16}$$

The minimum of $F$ may be attained on a cell of positive codimension, (see e.g. Fig. 1, left.) so restricting attention to full-dimensional cells is not sufficient. Among cells of positive codimension, there is no clear closed-form solution for a minimum.

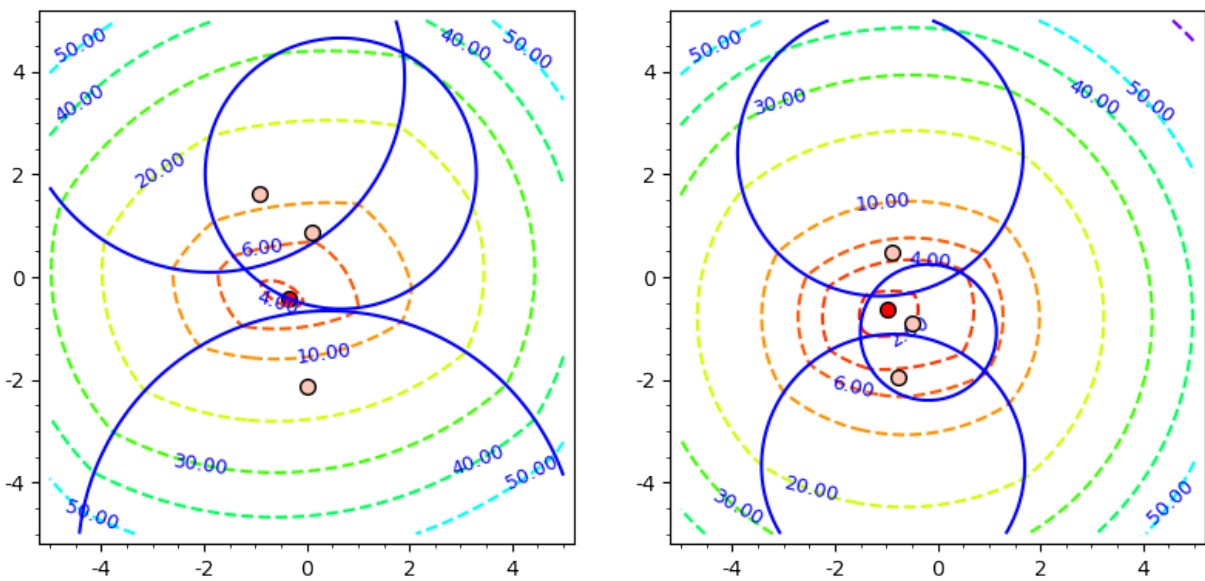

Figure 1: **Minima of $F$ on cells of various dimensions.** Data point in orange, minima in red. Selected level sets (in dotted-lines) are also reported.

## 4  Optimization: computing the unique minimizer of $F_\eta$

Having established the strict convexity of $F_\eta$, we compute its unique minimizer. (As in the previous section, we simply denote $F_\eta$ as $F$.) Our algorithm actually minimizes a function of the form $\sum_i \max(0, \|x - c_i\|^2 - R_i^2)$, and constructs a finite sequence of points $(x_n)$ by induction. The last point is the optimum of $F$. It assumes infinite precision – the so called *real RAM model*, and also assumes that all points $c_i$ are in a generic position and that when spheres $S_i$ intersect. See Sections 4.4 and 4.5 for comments on these assumptions.

### 4.1  Subdifferential and generalized gradient

We face a non-smooth convex optimization problem without constraint. Guarantees on the speed of convergence of algorithms with such assumptions as a number of iterations are rather weak. While gradients exists almost-everywhere, the classical gradient descent method may get trapped to bottleneck situations Bubeck (2015) leading to a precision rate of $O\left(\frac{1}{\sqrt{T}}\right)$, where $T$ is the number of steps. Further non-smooth investigation revolves around the use of the *subdifferential* or equivalently the *Clarke gradient* Clarke (1997) of the function. For a convex function, the subdifferential/Clarke gradient of $F$ at $x$, denoted by $\partial_* F(x)$ is a convex set defined below, while the generalized gradient $\nabla_* F(x)$ is its element of least norm (Fig. S2):

$$\begin{cases} \partial_* F(x) & := \{s, f(y) - f(x) \geq s \cdot (y - x), \forall y\} \\ \nabla_* F(x) & := \underset{u \in \partial_* F(x)}{\operatorname{argmin}} \|u\|. \end{cases} \tag{17}$$

Gradient samplings methods Burke et al. (2020) avoid the earlier described bottleneck configurations with a good descent direction obtained by approximating the generalized gradient. Given an arrangement of the space, the recent so-called stratified gradient sampling Leygonie et al. (2023) proposes to use the arrangement to efficiently determine a good descent direction. To tailor an exact algorithm, we use the structure at our disposal. Indeed, for any fixed $x$ in $\mathbb{R}^d$, we let $I^+, I^0$ and $I^-$ be the three-set partition associated to the cell of $x$. Then the subdifferential of $F$ at $x$ can be expressed as follows.

$$\partial_* F(x) = \{\nabla f_{I^+}(x) + \sum_{i \in I^0} \lambda_i \nabla f_i(x), 0 \leq \lambda_i \leq 1\}. \tag{18}$$

The generalized gradient can thus be expressed as a solution to the following quadratic programming (QP) problem, which admits a unique solution in $\lambda$ when all $c_i$ are in generic position – Sec. A:

$$\nabla_* F(x) = \underset{0 \leq \lambda_i \leq 1}{\operatorname{argmin}} \left\{ \|\nabla f_{I^+}(x) + \sum_{i \in I^0} \lambda_i \nabla f_i(x)\|^2 \right\}. \tag{19}$$

Letting $(\alpha_i)_{i \in I^0}$ be the unique solution to this problem, we let [1]

$$\begin{cases} I_*^0(x) & := \{i \in I^0(x), 0 < \alpha_i < 1\} \\ I_*^+(x) & := I^+(x) \cup \{i \in I^0(x), \alpha_i = 1\} \\ I_*^-(x) & := I^-(x) \cup \{i \in I^0(x), \alpha_i = 0\}. \end{cases} \tag{20}$$

Moreover, we let $\mathcal{C}^*(x)$ be the cell with three-set partition $I_*^+(x), I_*^0(x), I_*^-(x)$. This cell plays an important role in our algorithm, as the following paragraph about semiflows will demonstrate.

**Describing the semiflow of $F$.**  Even though $\nabla_* F$ might not be continuous, by convexity of $F$ from any starting point $x$ there exists (see for instance Bolte et al. (2010); Marcellin & Thibault (2006)) a trajectory $t \mapsto x(t)$ (with $x(0) = x$) called a *semiflow* verifying (for $t \in \mathbb{R}^+$):

$$x'(t) = -\nabla_* F(x(t)). \tag{21}$$

---

[1]Not to be confused with the sets $I^+(\mathcal{C}), I^0(\mathcal{C}), I^-(\mathcal{C})$, defined from the sign of the power distance.

In particular $F(x(t))$ decreases at rate $\|\nabla F_*(x(t))\|^2$, and by strong convexity $x(t)$ reaches the argmin of $F$ over $\mathbb{R}^d$ in a finite time, where it is stationary Marcellin & Thibault (2006); Josz (2023). Given the structure of our $F$, there are three possible behaviors for the semiflow with starting point $x$:

- If $x$ is in a full dimensional cell, the semiflow starting from $x$ begins by a segment heading towards $c_{I^+(x)}$, until it reaches a new cell or $c_{I^+(x)}$ which is the minimum.

- If $x$ lies in a cell of positive codimension and $I^0_*(x)$ is empty, the semiflow enters the non-empty, full dimensional cell $\mathcal{C}^*(x)$ and follows a straight line in this cell until it meets a new cell, as described as above.

- Else, $I^0_*(x)$ is not empty. One can show that if the Clarke QP (Eq. 19) lies in what we call a non-degenerate position[2], for small $t$ the semiflow enters the non-empty cell of positive codimension $\mathcal{C}^*(x)$, which is a subset of $S_{I^0_*(x)}$. Points $x$ with a degenerate QP problem are of measure zero, however points $x$ such that the trajectory $x(t)$ reaches a degenerate QP position, making the semiflow intractable, are not.

## 4.2 Exact algorithm

We develop algorithm `SC-Exact-Solver` mimicking the semiflow trajectory except for the third type of trajectory described above to seek for the minimum of $F$. See Algo. 5 for the pseudo-code – Sec. A.

It can be decomposed into three so-called main *procedures*, which are `Teleportation`, `LineDescent`, `SphereDescent`. The latter is further described using two procedures `SphereIntersection` and `MinSphereIntersection`. A sixth procedure used in `LineDescent` and `SphereDescent` is `ClarkeQP`. Except for the latter which consists in solving a classical QP programming problem, their pseudo-code can be found in – Sec A. Procedures `LineDescent`, `SphereDescent` are illustrated in Fig. 2.

(i) If $x_n$ lies in a full dimensional cell $\mathcal{C}$ (usually at the start of the algorithm), we check if $I^+(\mathcal{C})$ contains $c_{I^+(\mathcal{C})}$. If so, we let $x_{n+1}$ be $c_{I^+(\mathcal{C})}$ and we stop the algorithm. Since this step does not follow the semiflow we call it the `Teleportation` procedure. If there is no teleportation, we obtain $x_{n+1}$ from the `LineDescent` procedure within $\mathcal{C}$, which is described as follows. We seek the first point on the half-line starting from $x_n$ heading towards $c_{I^+(\mathcal{C})}$ meeting another cell, and we let $x_{n+1}$ be this point. This is done by solving for quadratic equations (in $t$) of the form $\|x_n + tu - c_i\|^2 = R_i^2$.

(ii) Else $x_n$ starts in cell of positive codimension. Compute the generalized gradient of $f$ at $x$ as well as the associated $I^+_*(x_n), I^0_*(x_n), I^-_*(x_n)$ with the `ClarkeQP` procedure (*i.e.* solving Eq. 19).

- If the generalized gradient is zero, the minimum has been reached and we can stop the algorithm.

- If $I^0_*(x_n)$ is empty, follow the `LineDescent` procedure described earlier within the full dimensional cell $\mathcal{C}^*(x_n)$. Take $x_{n+1}$ to be the point given by this procedure.

- Else, the semiflow starting from $x_n$ stays in $S_{I^0_*(x_n)}$ and we follow the `SphereDescent` procedure, which consists in the following. Compute the point $y$ where $f_{I^+_*(x_n)}$ restricted to $S_{I^0_*(x_n)}$ reaches its minimum via a procedure called `MinSphereIntersection` described in more details in the appendix. If $y$ is in the cell $\mathcal{C}^*$, let $x_{n+1}$ be $y$. Else, compute the center $c_S$ and radius $R_S$ of $S_{I^0_*(x_n)}$ via the `SphereIntersection` procedure. Via the parameterization $[0,1] \mapsto c_S + R_S \frac{(1-\lambda)x_n + \lambda y - c_S}{\|(1-\lambda)x_n + \lambda y - c_S\|}$ of the geodesic on $S_{I^0_*(x_n)}$, check the first point on the geodesic leaving the cell $\mathcal{C}^*$. Let $x_{n+1}$ be this point.

Following the semiflow ensures that our algorithm converges in a known number of steps in a certain neighborhood of the point $x^*$ where $F$ reaches its minimum. The number of steps is related to the number of

---

[2]We say that the QP problem of minimizing $\left\|u + \sum_i \lambda_i v_i\right\|, 0 \leq \lambda_i \leq 1$ lies in a non-degenerate position when the argmin $w$ is such that the set of $i$ such that $w \cdot v_i = 0$ is exactly the set of $i$ such that the coefficient of $v_i$ in the decomposition of $w$ is neither 0 or 1. This condition is standard in the sense that for a given box, for almost all isometries acting on that box, the image box lies in a non-degenerate position.

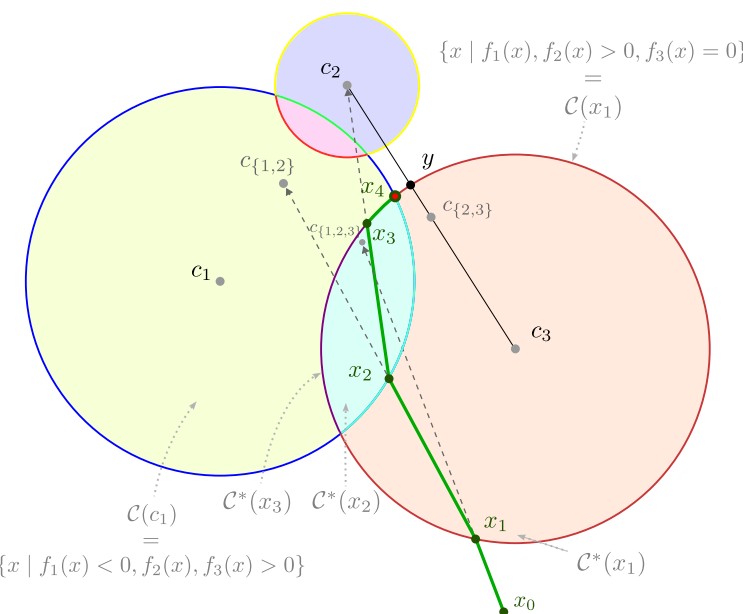

Figure 2: `LineDescent` **(from** $x_0, x_1, x_2$**) and** `SphereDescent` **(from** $x_3$**) steps. Underlying trajectories are depicted in dark green. Point** $y$ **is obtained by** `MinSphereIntersection` **point** $x_3$**.**

faces of the Clarke gradient $\partial_* F(x^*)$, which is $3^c$ where $c$ is the number of spheres on which $x^*$ lies. We refer the reader to the proof in appendix for the quantification of the neighborhood size.

**Theorem. 2** (Algorithm convergence)**.** *Denote by $x^*$ the point where $F$ reaches its minimum, and let $c$ be the number of spheres on which $x^*$ lies. There exists a ball centered at $x^*$ with radius $R > 0$ depending on (i) the maximum radius of the ball only cells touching $x^*$, and (ii) the Clarke gradient $\partial_* F(x^*)$, in which the algorithm converges in at most $3^c$ steps.*

### 4.3 Combinatorial complexity

The complexity (number of cells of all dimensions) of the arrangement of $n$ spheres in $\mathbb{R}^d$ is $O(n^d)$ Toth et al. (2017) and the bound is tight in the worst case. Despite this, we may expect a number of steps polynomial in $n$ if calculations remain *local* in the arrangement, a fact substantiated by our experiments. Note that for small values of $\eta$, the center of mass provides a warm start to the algorithm. Indeed the solution of the minimization problem of $F_\eta$ varies continuously in $\eta$, and the barycenter is solution to the problem with $\eta = 0$.

Computations at each step are linear in the total number of points since `LineDescent` (resp. `SphereDescent`) computes the first sphere crossed by a line (resp. a sphere geodesics). Other computations involved at each step are at most cubic in the number of spheres on which the current point lies, be it by solving the `ClarkeQP` problem, inverting a matrix in `SphereIntersection` or computing an affine projection in `MinSphereIntersection`.

While the limiting factor of our approach is the absence of constructive bound on the number of steps, we point out that contrary to classical methods, our theoretical analysis shows that the exact minimizer is reached in a finite number of steps, and that this number is bounded when the algorithm starts in a neighborhood of a certain size (see Theorem 2 for the exact statement). In classical methods such as the subgradient descent, the final point is guaranteed to lie at distance at most $\varepsilon$ to the minimizer after a number of steps tending to infinity as $\varepsilon$ goes to 0. To ensure an exact convergence with precise complexity guarantees, one could thus use a subgradient descent and finish the job with our algorithm. In practice, we did not have to resort to such ad-hoc methods, as the algorithm outperforms the classical methods used in non-smooth convex optimization – see Sec. 5.5.

### 4.4 Numerics

**Arithmetics and number types.** The algorithm from Sec. 4 is described assuming the real RAM model computing exactly with real numbers. On the other hand, geometric calculations (predicates, constructions) are known to be plagued with rounding errors Kettner et al. (2008). Serious difficulties may be faced for cascaded constructions, which iteratively embed new geometric objects (the points of the pseudo-gradient trajectory in our case). Advanced number types combining multiprecision and interval arithmetics can be used to maintain accurate representations in such cases. See *e.g.* random walk inside polytopes Chevallier et al. (2022) or trajectories of the flow complex (the Morse-Smale diagram of the distance function to a finite point set) Cazals et al. (2021).

In the sequel, we review the numerically demanding operations required by our algorithm, and refer the reader to Sec. 5 for experiments with our python based implementation.

**Exact solver.** The solver uses the following predicates and constructions:
•**Solving the** `ClarkeQP` **procedure.** The library *cvxpy* uses non-exact solvers to find the minimum of a QP problem. While those solvers usually give a result up to machine precision, we found out that they were prone to instability in minimizing functionals of the type $\|g + A\lambda\|^2$ when $g$ is vector of norm largely greater than both 1 and than that of the columns of $A$, with constraints $0 \preceq \lambda \preceq 1$, in the sense that those solvers would claim the problem to be unfeasible. The equivalent problem of minimizing $\left\|\frac{1}{\|g\|}(g + A\lambda)\right\|^2$ was sufficient in addressing those issues. The precise computation of the vector $\lambda$ is not needed as we only need to check for the index $i$ with respectively $\lambda_i \in \{0\}, (0,1)$ and $\{1\}$. The entries with values 0 and 1 are usually reached with precision greater than machine precision.

•**Solving the** `LineDescent` **and** `SphereDescent` **procedures.** Starting from a point $x$, with a prescribed direction $u$, procedure `LineDescent` seeks the first $t$ such $x + tu$ changes cell, that is, the smallest positive $t$ verifying $\|x + tu - c_i\|^2 = R_i^2$. This is obtained as roots of a second-degree polynomial. To weaken imprecision we chose to solve this equation with a renormalized vector $u$ of norm 1. Similarly, given two points $x, y$ on a sphere of radius $R_S$ and center $c_S$, procedure `SphereDescent` computes the first point on the geodesic between $x$ and $y$ changing cell by solving for a quadratic equation.

•**Solving the** `SphereIntersection` **procedure.** The pair center/radius $c_S, R_S$ used above is obtained as the center and radius of an intersection of spheres. As described in the appendix, the center is obtained through the computations of a projection onto an hyperplane defined by linear equations involving the centers of said spheres. Computing $R_i$ is done by solving for a quadratic equation.

•**Solving for the** `MinSphereIntersection` **procedure.** Given an intersection of spheres $S_I$, with $I \subset \{1, \ldots, n\}, \#I \leq d$ the `MinSphereIntersection` procedure minimizes a function of the form $(f_J)_{|S_I}$ by computing a similar projection on a convex hull.

The genericity assumptions and the robustness of our routines are further discussed in the SI Section 4.5.

**The BFGS solver.** The Broyden–Fletcher–Goldfarb–Shanno quasi-Newton method (BFGS) is designed to be very efficient on twice differentiable function by approximating the Hessian matrix without any matrix inversion (in opposition to Newton's methods), using the gradient. When the gradient is not given, it is estimated using finite differences. While the objective function $F_\eta$ is not differentiable, it is also known that BFGS works well in practice for non-differentiable functions Lewis & Overton (2012). The next section challenges this observation for $F_\eta$.

The BFGS solver is systematically launched from a warm start at the center of mass of the point cloud processed – as for the exact solver.

### 4.5 Genericity assumptions

The genericity condition for our algorithm to work at a point $x$ is that the intersection of spheres containing $x$ is transverse. For $k$ spheres, this existence of a non transverse intersection is checked as follows. The non transversality at point $x$ reads as $\sum_i \lambda_i(x - c_i) = 0$ or equivalently $(\sum_i \lambda_i)x = \sum_i \lambda_i c_i$, which requires

discussing two cases: (Case 1) $\sum_i \lambda_i \neq 0$, and (Case 2) $\sum_i \lambda_i = 0$. In each case, we need to check the points $x$ satisfying these conditions satisfy the sphere equations.

To do so, Case 1 requires solving a QP problem. Case 2 requires computing the null space of the matrix $A = (\{[c_i \ 1]^T\}_{i=1,\dots,k})$, and if $Ker(A) \neq 0$, one needs to further check the sphere feasibility conditions, which require solving another linear system.

Also note that at a given point $x$, the test simply boils down to checking that the vectors $x - c_i$ are linearly independent.

The exact computation of trajectories by our algorithm is more involved. It requires cascaded degree two and degree four algebraic numbers. (NB: degree two when intersecting a segment with a sphere; degree four when intersecting a geodesic along a sphere with another sphere.) A robust numerical solution could be obtained using say an interval number type with bounds of arbitrary precision, e.g. the iRRAM library Müller (2001).

In practice though:

- We do not check the transversality condition, as even in medium dimensional spaces, the points where the intersections are not transverse are scarce, and our trajectories do not cross them.

- We do not use elaborate number types, since the observed robustness of our floating point implementation did not require using them.

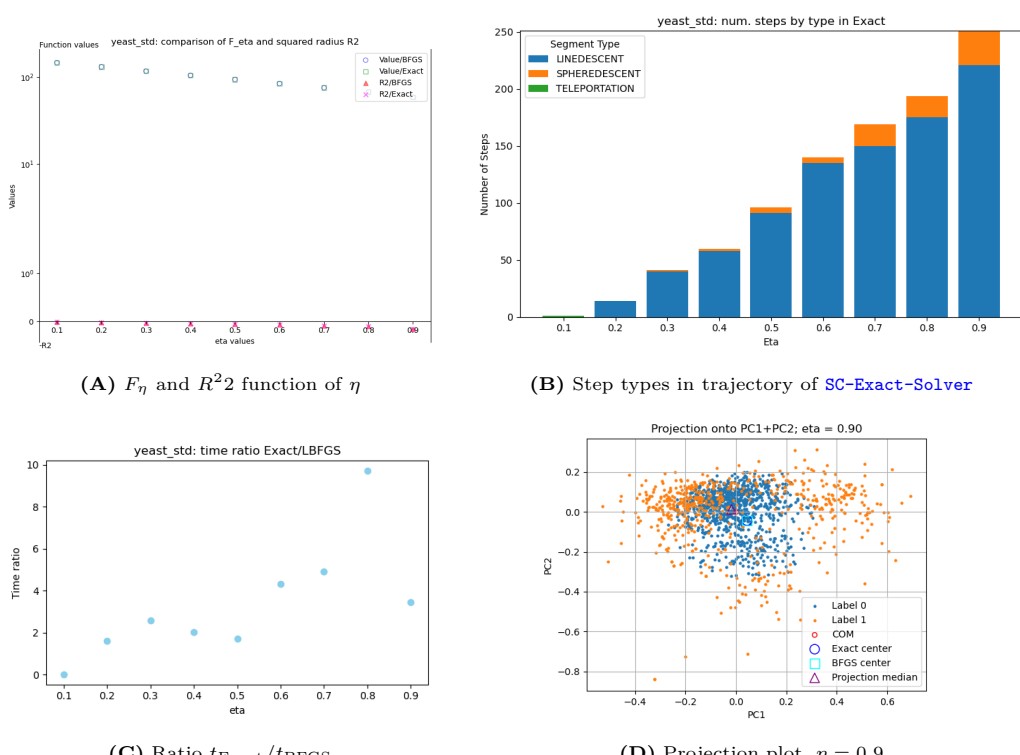

**(A)** $F_\eta$ and $R^2 2$ function of $\eta$

**(B)** Step types in trajectory of SC-Exact-Solver

**(C)** Ratio $t_{\text{Exact}}/t_{\text{BFGS}}$

**(D)** Projection plot, $\eta = 0.9$

Figure 3: **Yeast landsat.** This dataset features 6435 points in dimension $d = 9$. **(A)** Dual plot **(B)** Steps types as a function of $\eta$ **(C)** Running times $t_{\text{Exact}}$ vs $t_{\text{L-BFGS-B}}$ **(D)** Projection plot with inliers and outliers

# 5 Spherical clusters: experiments

## 5.1 Implementation

Our implementation of the algorithm from Sec. 4 using python and numpy is denoted `SC-Exact-Solver` and is termed the *exact method*. It is available from the Core tier of the Structural Bioinformatics Library, in the Cluster spherical package. We compare the solution yielded by `SC-Exact-Solver` against that yielded by `SC-BFGS-Solver`– the optimization being done with BFGS. The cluster centers are denoted $\text{opt}_{\text{Exact}}$ and $\text{opt}_{\text{BFGS}}$ respectively.

Calculations were run on a DELL precision 5480 equipped with 20 CPUs of type Intel(R) Core(TM) i9-13900H, 32Go or RAM, and running FedoraCore 42.

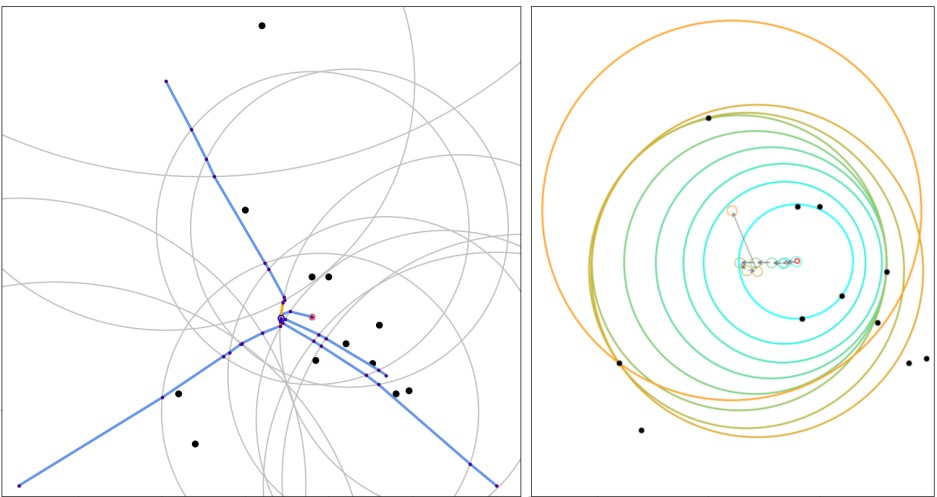

Figure 4: **Spherical cluster: illustrations on a toy 2D dataset. (Left)** Trajectories from five different starting points, with $\eta = 0.5$ (Line/Sphere descents in blue/orange). **(Right)** Evolution of the cluster center for $\eta$ in $[0.1, 0.9]$ be step of 0.1.

## 5.2 Contenders, datasets and statistics

**Contenders.** We challenge `SC-Exact-Solver` with two contenders denoted `SC-BFGS-Solver` and `SC-LBFGS-Solver` respectively, using the BFGS and L-BFGS-B solvers provided by `scipy.optimize`. Note that the latter uses an approximation of the Hessian–as opposed to a $O(d^2)$ sized matrix.

**Medium dimensional (MD) datasets.** We ran experiments on ten standards datasets used in clustering experiments Celebi et al. (2013); Carrière & Cazals (2025), with size $n \in [1484, 200000]$ and dimension $d \in [9, 77]$ – Table S1. Following common practice, on a per dataset basis, we perform a min-max normalization on the coordinates to avoid overly large ranges.

**High dimensional (HD) datasets.** We use two datasets to explore the effect of high dimensionality. The `Proteins-HMM` dataset consists of $N = 1443$ protein sequences whose biological function is unknown Vicedomini et al. (2022). To identify putative functions, each sequence is scored by $d = 400$ Hidden Markov Models (HMM) corresponding to major known protein functions, yielding a $d$-dimensional point. Carbone et al. perform hierarchical clustering on these points (Ward's method), yielding 16 clusters (sizes in 11..176) of sequences expected to have identical functions. The Arcene dataset contains mass-spectrometric data meant to distinguish cancer versus normal patients, and has shape $(n, d) = (900, 10000)$. The $d = 10000$ features correspond to protein abundances in human sera, to which distractor features with no predictive power have been added.

**Parameters.** For each dataset, we explore values of $\eta$ in $[0.1, 0.9]$ by steps of $0.1$ – nine values in total.

**Statistics and plots.** We define (Fig. 3 and SI):

•*Spherical cluster square radius $R^2$.* The square radius with respect to which the power distance is computed, that is $\eta\hat{\sigma}^2$ – see Eq. 1.

•*Projection plot.* The plot of all points (data points, center of mass, SC centers) onto the first two principal directions. Inliers (resp. outliers) are displayed in blue (resp. orange). The number of outliers identified by our cluster model, is denoted #outliers(SC). Similarly #outliers(COM) stand for the number be outliers defined with respect to a sphere of the same radius centered at the center of mass.

•**Dual plot.** Reports $F_\eta$ and $R^2$ as a function of $\eta$. (NB: $R^2$ values are represented negated on this plot.)

•**Stacked barplot.** The plot function of $\eta$ counting the number of steps of each type (line descent, sphere descent, teleportation) in `SC-Exact-Solver`.

•**Time ratio plots.** The plots for $t_{\text{Exact}}/t_{\text{BFGS}}$ and $t_{\text{Exact}}/t_{\text{L-BFGS-B}}$, comparing the running times of `SC-Exact-Solver` against those of `SC-BFGS-Solver` and `SC-LBFGS-Solver` respectively.

•**Average outlier cost plots.** The plots $F_\eta(\text{opt}_{\text{Exact}})/\text{#outliers(SC)}$ and $F_\eta(\text{opt}_{\text{Exact}})/\text{#outliers(COM)}$.

•**Outlier ratio plot.** The plot #outliers(COM)/#outliers(SC).

•**Distance between points plot.** The plot comparing the distances between three special points: $\text{opt}_{\text{Exact}}$, $\text{opt}_{\text{BFGS}}$, and the projection median from Durocher et al. (2017).

## 5.3 Spherical cluster model

**Trajectories and centers.** We build up an intuition by observing the trajectories followed by our exact solver when varying the starting point, on a simple toy 2D example (Fig. 4(A)). We also note that even for such simple cases, the center moves in a complex way when varying $\eta$ (Fig. 4(B)).

**Running times and the burden of dimensionality.** For the dataset `DS-Cluster`, we first inspect running times using the ratios $t_{\text{Exact}}/t_{\text{L-BFGS-B}}$ (Fig. S5, Tab. S3), and $t_{\text{Exact}}/t_{\text{L-BFGS-B}}$ (Fig. S6, Tab. S5). Using median values, the comparison shows that `SC-Exact-Solver` is faster than `SC-BFGS-Solver` up to up to $\eta = 0.7$ included, while `SC-Exact-Solver` is faster than `SC-LBFGS-Solver` up to $\eta = 0.3$ included. Increasing the value of $\eta$ results in larger spheres and more complex arrangements, whence the burden observed.

We perform the same analysis for the dataset `Proteins-HMM`, for ratios $t_{\text{Exact}}/t_{\text{L-BFGS-B}}$ (Fig. S11, Tab. S7), and $t_{\text{Exact}}/t_{\text{L-BFGS-B}}$ (Fig. S12, Tab. S9). Using median values again, `SC-Exact-Solver` is two for four orders of magnitude faster than `SC-BFGS-Solver` and `SC-LBFGS-Solver`.

For the Arcene dataset, BFGS turned out to be unpractical. We observe that our exact algorithm is between two and five orders of magnitude faster than `SC-LBFGS-Solver` (Fig. S17).

Summarizing, `SC-Exact-Solver` is orders of magnitude faster than `SC-LBFGS-Solver` and `SC-LBFGS-Solver` for datasets of small/intermediate dimension and small values of $\eta$, and orders of magnitude faster than these two methods for high dimensional datasets.

**Function values.** Wen now compare the values yielded by the three contenders: `SC-Exact-Solver` vs `SC-BFGS-Solver`: Fig. S11 and Table S8; `SC-Exact-Solver` vs `SC-LBFGS-Solver`: Fig. S12 and Table S10. While these values are on par for all values of $\eta$, we note that the approximate solvers are more prone to numerical instabilities, in particular for `DS-Cluster` and for `DS-HD`.

**Outliers and the selection of $\eta$.** As noticed earlier, the SC center depends both on inliers and outliers. On all datasets processed, the outlier ratio #outliers(COM)/#outliers(SC) lies in the interval $\sim [1, 3]$, which illustrates the stringency of our criterion to identify such points.

The outlier cost plot $F_\eta(\text{opt}_{\text{Exact}})/\#\text{outliers}(SC)$ is of particular interest to capture the scale/cost of outliers. The general behavior of this plot is a monotonic decrease (*e.g.* Fig. S13, Fig. S15), indicating that *capturing* outliers is getting easier when increasing $\eta$. However several datasets exhibit a non monotonic behavior (Fig. S7, Fig. S9), showing that *gaps* must be crossed to capture certain outliers.

### 5.4 Projection median

We also compare $\text{opt}_{\text{Exact}}$ against the projection median computed as a weighted average Durocher et al. (2017). As expected, their distance increases as a function of $\eta$, showing that the cluster center behaves as a parameterized point set center. See Supporting Information, plots *Distance between points* plots.

### 5.5 Discussion: complexity in practice

**Number of steps and multiplicity of cells.**   For all datasets and whatever the value of eta, we checked that the number of cells traversed is negligible with respect to the worst case complexity of the arrangement. We also checked that in practice the cells visited are so only once, contrary to what can be found in pathological sphere configurations with bad starting points. We draw a comparison with the celebrated simplex algorithm, which has exponential complexity in the worst case, and yet stays in use after almost 80 years. Our framework is similar in spirit, as the arrangement is exponential in dimension while the algorithm is effective in practice. Note that we were unable to build an example where the number of steps is anywhere near the total number of cells. The fact that our trajectory benefits from a *warm start* (warmer as eta decreases) at the center of mass certainly helps in reducing the number of cells to be crossed before reaching the minimum, explaining the efficiency of our algorithm against classical methods (whose underlying trajectory is not stopped when crossing a cell) when eta is not close to $1 - 1/n$.

**Behavior in high-dimensions.**   In higher dimensions, our algorithm outperforms the BFGS and L-BFGS. The main factor for this behavior resides in computation times of the steps which are cubic in the number of spheres containing the current point of the trajectory–see Section 4.3). In practice, this number is small, on average between 2 and 3 as shown by our experiments: a vast majority of steps are either `LineDescent` procedures or `SphereDescent` on a small number of spheres–rarely more than five even in high dimensions.

## 6 Outlook

Spherical clusters embedded in affine spaces of fixed dimension provide useful insights into the geometry of high dimensional point clouds.

Our work shows spherical clusters are well defined by a non smooth strictly convex problem. We also show that this optimization problem is well poised and can be solved by an exact iterative procedure following a semiflow on a stratified complex defined by an arrangement of spheres. Quite remarkably, BFGS also solves all the instances we processed to satisfaction. Yet, the exact solver is orders of magnitude faster than BFGS based heuristics for high dimensional datasets (say $d > 100$), and for dataset of medium dimensionality and small values of $\eta$. Our experiments also show that the center of spherical clusters behave as a high dimensional median parameterized by the fraction $\eta$ of the variance of distances between the cluster center and all points.

Our work calls for future developments in theory and in practice.

From a theoretical standpoint, understanding the complexity of our exact method as a function of $\eta$ appears as a challenging problem.

From a practical standpoint, spherical and affine clusters were proposed as mixtures components. However, fitting such mixtures is a challenging non convex problem which commands to monitor the quality of the fit and the model complexity. To the best of our knowledge, two main strategies have been explored for this task. The first one is based on split/merge/delete operations on components of the mixture, a very demanding task Kasarapu & Allison (2015). The second one consists of Expectation-Maximization based strategies Dempster et al. (1977), possibly combined with model control using *e.g.* the minimum message

length Figueiredo & Jain (2002). This is also a demanding strategy, in particular to control the singularization of the components and their number.

Defining spherical clusters embedded into affine spaces of the varying dimensionality appears as a very appealing choice, but a non trivial task. If successful, we anticipate that such models will prove extremely useful in data analysis at large, providing compact clusters capturing the intrinsic dimension of the data, that could also be used to define stratified complexes.

**Acknowledgements**

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
