# OpenReview forum: "Modeling high dimensional point clouds with the spherical cluster model"
_TMLR — Rejected by TMLR_

### Review · Reviewer_XF2L · 2026-04-02

**Summary Of Contributions:**

This work considers spherical clusters of points in Euclidean space. It shows that fitting such a spherical cluster is a strictly convex problem and presents an algorithm to do so. Under the assumption of infinite precision and generically positioned points, a convergence guarantee for the algorithm is derived. If the starting point is in a certain (not further specified) neighborhood of the minimum, then the algorithm converges, and moreover an upper bound of the necessary number of steps is known.
Empirically, the runtime and cluster quality is compared with is compared with BFGS solvers.

**Audience:**

Yes

**Audience Explanation:**

Clustering is one of the most commonly used machine learning and data analysis techniques. Therefore, this work is certainly of interest to at least some individuals in TMLR's audience.
Specifically, the method's applicability to high dimensional data might be of interest.

**Broader Impact Concerns:**

No concerns

**Claims And Evidence:**

Yes

**Claims Explanation:**

Unfortunately I don't have sufficient background on algorithms to adequately assess the correctness of its convergence guarantee (Theorem 2).

Regarding the empirical evaluation, runtime is reported as a fractions of the runtime of the exact solver and BFGS. While this certainly demonstrates the benefits of the exact solver over BFGS (it is multiple orders of magnitude faster), it does not provide information about the practicality of the method, i.e., the exact solver could still be impractically slow, especially on the high dimensional datasets (The title of the work suggests that the method is designed for high dimensional data). Additionally reporting absolute runtime would provide more clarity. This point could be further strengthened if more than 2 high dimensional datasets are used.

**Requested Changes:**

- Report absolute runtime on all datasets

Minor (would simply strengthen the work)

- Theorem: Characterize the neighborhood of the point $x^*$ precisely. Looking at the proof, it is clear that the "there exists a neighborhood" is not only an existential statement but also constructive.
- Clarify the complexity discussion. How is the number of cells of all dimensions related to the number of  steps, (which depends on the number of spheres in which $x^*$ lies$
- Clarify the term sphere in contrast to balls or disks. Particularly, in theorem 2, is it the number of spheres *inside* of which or *on* which $x^*$ lies.
- Discuss how the technique would be used / adapted if one wants to fit multiple clusters to data
- Improve figure quality: Text and markers in the figures are often quite small, making it difficult to extract information. In some plots, the axis range and scaling is to coarse, e.g. Figure 14 right.
- Equation 1: While related to variance of a cluster, this is not the "unbiased variance estimate for distances within cluster $D_l$".
- Definition 2: "be is"

---

> ### Author Response · Authors · 2026-04-28
> **Complements to our general answer.**
>
> ## QUESTION.
>
> Theorem: Characterize the neighborhood of the point $x^*$ precisely.
>
> ## ANSWER.
>
> The size of the neighborhood has been explicit stated, and the corresponding proof has been made clearer, notably with the additions of two figures.
>
> ## QUESTION.
>
> Clarify the complexity discussion -- number of cells and complexity.
>
> ## ANSWER.
>
> As explained above, we have clarified the final convergence guarantee in the main theorem.
>
> ## QUESTION.
>
> Clarify the term sphere in contrast to balls or disks.
>
> ## ANSWER.
>
> We added one sentence stating that a ball is bounded by a sphere.
>
> ## QUESTION.
>
> Discuss how the technique would be used / adapted if one wants to fit multiple clusters to data.
>
> ## ANSWER.
>
> This topic was touched upon in the Introduction.
> We enlarged the paragraph in Introduction, and also commented
> on mixtures of spherical clusters embedded into affine spaces in the Outlook.
>
> This problem is as interesting as challenging. We believe it deserved a full paper in itself.
>
>
> ## QUESTION.
>
> Improve figure quality: Text and markers in the figures are often quite small, making it difficult to extract information. In some plots, the axis range and scaling is to coarse, e.g. Figure 14 right.
>
>
> ## ANSWER.
>
> The figures presenting the 9 projection plots were indeed hard
> to read. We moved from 9 to 6 projection plots per model, favoring
> high values of $\eta$ -- since the optimization problem is simpler for
> small values of $\eta$.  Note though that the remaining figures use a
> spacing of 0.1 in $\eta$ space.
>
> ## QUESTION.
>
> While related to variance of a cluster, this is   not the unbiased variance estimate for distances within
>   cluster $D_l$.
>
> # # ANSWER.
>
> Correct. We renamed it the *sample variance*.

---

### Review · Reviewer_W2hj · 2026-04-04

**Summary Of Contributions:**

The paper studies the single-center fitting problem for one given set of points. Instead of using the usual centroid, the authors define an objective where a candidate center induces a sphere whose radius depends on the spread of the data around that center, and only points outside that sphere are penalized. The main technical contribution is to show that this objective is nonsmooth but still strongly/strictly convex, so the solution is unique. On top of that, the paper develops a specialized exact solver based on Clarke generalized gradients and the arrangement of spheres induced by the objective. The experiments then show that this exact solver is often much faster than generic BFGS/L-BFGS-B baselines, especially in higher dimensions, and that the resulting center behaves somewhat like a tunable high-dimensional median.

Strengths:
* The paper shows that the proposed center is well defined and unique, despite the non-smooth objective.
* The method appears able to solve this objective much faster than generic quasi-Newton baselines on the tested datasets, especially in high dimensions.
* The resulting center provides a median-like alternative to the centroid for summarizing a high-dimensional point cloud.

Weaknesses:
* The main weakness of the paper is that the motivation is not clear. The paper is presented as clustering, but the main object here is really one center for one cloud.
* The objective is the paper’s own creature. And, again, it's not clear why this particular objective is worth studying.
* The experiments only validate the solver. And, again, it's not clear how downstream ML applications can benefit from the solver.

**Audience:**

Yes

**Audience Explanation:**

This paper would be of interest to at least a subset of the TMLR audience, especially readers working on clustering, robust center estimation, high-dimensional geometry, or non-smooth optimization. The main caveat is that the paper feels more like a technical optimization / geometry contribution than a broadly impactful ML paper, so I expect the interested audience to be somewhat specialized rather than wide.

**Broader Impact Concerns:**

No concern.

**Claims And Evidence:**

Yes

**Claims Explanation:**

The main theoretical and algorithmic claims are supported reasonably well by the paper and the supplement. In particular, the analysis and proofs support the claims about the structure of the objective, uniqueness of the minimizer, and the proposed solver. The experiments also support the claim that the solver is often faster than BFGS/L-BFGS-B on the tested datasets. My main reservation is that the evidence is much stronger for the optimization results than for the broader usefulness of the model itself.

**Requested Changes:**

Critical:
* Clarify the motivation. Right now the paper is framed as clustering, but the actual problem studied here is fitting a single center to a single point cloud. The paper should explain more directly why this problem matters on its own, and why this particular spherical-cluster objective is worth studying.
* Better justify the modeling choice. Since the objective is largely the paper’s own creature, the paper should explain why and how this radius definition and truncation form are natural and intuitive, and why this should be preferred over other robust center estimators.
* Improve the presentation of contributions. The paper should be more explicit that the main result here is a single-center estimator plus an exact solver, not a full clustering method. Right now the framing risks overstating the scope.
* State the admissible range of $\eta$ consistently. The paper introduces $\eta\in(0,1)$, but the later analysis appears to require the stricter condition $0<\eta<1-\frac{1}{n}$. This should be stated clearly and consistently throughout.
* Tighten the mathematical description of the radius term. The paper describes the radius as being based on the “standard deviation” or “variance of distances,” but the actual quantity used is $\hat{\sigma}^2(c)=\frac{1}{n-1}\sum_i \|x_i-c\|^2$, i.e. an average squared distance to the candidate center rather than the usual variance of the scalar distances $\|x_i-c\|$. The wording should be made consistent with the formula.
* Clarify the scope of the convergence guarantee. The optimization problem appears globally well posed, but the stated convergence theorem for the proposed solver is local. The paper should make this distinction explicit and avoid giving the impression of a stronger global guarantee than what is proved.

Good to have:
* Make the ML relevance more explicit. As written, the paper reads more like a computational geometry / nonsmooth optimization paper. The authors should do a better job connecting the contribution to problems that ML readers would actually care about.
* Expand the experimental comparison beyond BFGS/L-BFGS-B. Those baselines make sense for the solver story, but they do not say much about the value of the model itself. Comparisons to other robust center estimators would help.
* Give a clearer practical interpretation of the parameter \eta. The paper shows how the center changes with \eta, but it is still not very clear how one should choose it in practice.
* Tighten the writing and polish. There are places where the wording is a bit awkward or inconsistent, and the abstract/introduction could be clearer.
* Clarify the role of the supplementary material. A lot of important proof and algorithm details are deferred there. That is fine, but the main paper could do a slightly better job surfacing the most important takeaways.

---

> ### Author Response · Authors · 2026-04-28
> **Complements to our general answer.**
>
> ## QUESTION.
>
> Weaknesses.  The main weakness of the paper is that the
> motivation is not clear. The paper is presented as clustering, but the
> main object here is really one center for one cloud.  The objective is
> the paper’s own creature. And, again, it's not clear why this
> particular objective is worth studying.  The experiments only validate
> the solver. And, again, it's not clear how downstream ML applications
> can benefit from the solver.
>
> ## ANSWER.
>
> This calls for three comments:
>
> * We disagree with the claim *The paper is presented as clustering*: the title, the Abstract and the Contributions
> *do not mention* any novel clustering contribution. Instead,
> we focus on the spherical cluster model.
>
> * We also disagree with the statement *The objective is the paper’s
> own creature*: we solve a problem introduced in the 2009 paper by Wang
> et al.  This is correctly pointed out in the second review, where one
> reads *This paper addresses a leftover of the spherical cluster (SC)
> model regarding the uniqueness of the center, and proposes new
> optimization algorithm to solve SC.*
>
> * Our experiments do not just validate the solver. Consider the
>   statistics mentioned as the end of section 5.2 of the submission,
>   namely *average outlier cost*, *outlier ratio* (wrt COM),
>   and *Distance between points* (wrt the projection median).  These
>   statistics provide insights on the behavior of our centerpoint with
>   respect to the center of mass and the projection median.
>
>
> ## QUESTION.
>
> Critical/Clarify the motivation.
>
> ## ANSWER.
>
> We hope this has been done in the new Section 2.3, whose conclusion
> mentions the merits of the spherical cluster model.
>
> ## QUESTION.
>
> Critical/Better justify the modeling choice.
>
> ## ANSWER.
>
> We hope this has been done in the new Section 2.3.
>
> ## QUESTION.
>
> Critical/Improve the presentation of contributions.
>
> ## ANSWER.
>
> We hope the revised Introduction, and in particular the new section
> *Spherical clusters: discussion* clarify our contributions -- and
> the difficulties to derive them.
>
>
> ## QUESTION.
>
> Critical/State the admissible range of $\eta$ consistently.
>
> ## ANSWER.
>
> Done.
>
> ## QUESTION.
>
> Critical/Tighten the mathematical description of the radius term.
>
> ## ANSWER.
>
> Done in section 2.3.
>
> ## QUESTION.
>
> Critical/Clarify the scope of the convergence guarantee.
>
> ## ANSWER.
>
> We clarified the proof for the convergence of the algorithm, in particular expliciting the size of the neighborhood of the minimizer in which our algorithm converges in a finite number of steps.
>
> ## QUESTION.
>
> Good to have. Make the ML relevance more explicit.
>
> ## ANSWER.
>
> We hope the revamped Introduction which positions the work
> at the cross-roads of clustering, cluster models, and center points, provides
> a clear positioning in the realm of ML.
>
> ## QUESTION.
>
> Good to have.  Expand the experimental comparison beyond BFGS/L-BFGS-B. Those baselines make sense for the solver story, but they do not say much about the value of the model itself. Comparisons to other robust center estimators would help.
>
> ## ANSWER.
>
> It is well admitted that solvers using sub-gradient descents are slower than L-BFGS-B,
> especially in high dimension, which is why we focused on BFGS/L-BFGS-B.
>
> About the value of the model itself, see the next point.
>
> ## QUESTION.
>
> Good to have. Give a clearer practical interpretation of the parameter $\eta$.
>
> ## ANSWER.
>
> See answer above about the following statistics:
> *average outlier cost*, *outlier ratio* (wrt COM),
> and *Distance between points (wrt the projection median).
>
> ## QUESTION.
>
> Good to have. Tighten the writing and polish.
>
> ## ANSWER.
>
> The paper contained some typos we corrected and a few ambiguous phrasings, which we rewrote. We also clarified some proofs.
>
> ## QUESTION.
>
> Good to have. Clarify the role of the supplementary material.
>
> ## ANSWER.
>
> We opted for a presentation striking a balance between (i) the problem, (ii)
> the mathematics properties of the functional, and (iii) experiments.
> We believe that moving most of the technical content into the main text
> would be detrimental for most readers -- those not versed into non smooth
> optimization.

---

> ### Comment · Reviewer_W2hj · 2026-04-30
> **I will not review the edits in the paper.**
>
> Please address my comments and concerns here with pairs of question/concern and answer/edit. I cannot go back and forth between the revised paper and my comments here to find which edit is addressing which of my comments.

---

> > ### Author Response · Authors · 2026-05-02
> > **Update of answers with reference to section for each QUESTION/ANSWER**
> >
> > **As per your request, please find below (i) a structured answer and copy/paste/summary of your question,and (ii) the answer plus a reference to the section(s) if applicable.**
> >
> > ## QUESTION.
> >
> > Weaknesses.  The main weakness of the paper is that the
> > motivation is not clear. The paper is presented as clustering, but the
> > main object here is really one center for one cloud.  The objective is
> > the paper’s own creature. And, again, it's not clear why this
> > particular objective is worth studying.  The experiments only validate
> > the solver. And, again, it's not clear how downstream ML applications
> > can benefit from the solver.
> >
> > ## ANSWER.
> >
> > This calls for three comments:
> >
> > * We disagree with the claim *The paper is presented as clustering*: the title, the Abstract and the Contributions
> > *do not mention* any novel clustering contribution. Instead,
> > we focus on the spherical cluster model.
> >
> > * We also disagree with the statement *The objective is the paper’s
> > own creature*: we solve a problem introduced in the 2009 paper by Wang
> > et al.  This is correctly pointed out in the second review, where one
> > reads *This paper addresses a leftover of the spherical cluster (SC)
> > model regarding the uniqueness of the center, and proposes new
> > optimization algorithm to solve SC.*
> >
> > * Our experiments do not just validate the solver. Consider the
> >   statistics mentioned as the end of section 5.2 of the submission,
> >   namely *average outlier cost*, *outlier ratio* (wrt COM),
> >   and *Distance between points* (wrt the projection median).  These
> >   statistics provide insights on the behavior of our centerpoint with
> >   respect to the center of mass and the projection median.
> >
> >
> > ## QUESTION.
> >
> > Critical/Clarify the motivation.
> >
> >
> > ## ANSWER.
> >
> > We hope this has been done in the new Section 2.3, whose conclusion
> > mentions the merits of the spherical cluster model.
> >
> > Update(s): Section 1/Cluster models.
> >
> > ## QUESTION.
> >
> > Critical/Better justify the modeling choice.
> >
> > ## ANSWER.
> >
> > We hope this has been done in the new Section 2.3.
> >
> > Update(s): Section 2.3.
> >
> > ## QUESTION.
> >
> > Critical/Improve the presentation of contributions.
> >
> > ## ANSWER.
> >
> > We hope the revised Introduction, and in particular the new section
> > *Spherical clusters: discussion* clarify our contributions -- and
> > the difficulties to derive them.
> >
> > Update(s): end of Section 1.
> >
> > ## QUESTION.
> >
> > Critical/State the admissible range of $\eta$ consistently.
> >
> > ## ANSWER.
> >
> > Done.
> >
> > Update(s): Def. 2.
> >
> > ## QUESTION.
> >
> > Critical/Tighten the mathematical description of the radius term.
> >
> > ## ANSWER.
> >
> > Done in section 2.3.
> >
> > Update(s): sentence before Def 1 + Section 2.3
> >
> > ## QUESTION.
> >
> > Critical/Clarify the scope of the convergence guarantee.
> >
> > ## ANSWER.
> >
> > We clarified the proof for the convergence of the algorithm, in particular expliciting the size of the neighborhood of the minimizer in which our algorithm converges in a finite number of steps.
> >
> > Update(s): statement of Thm 2 + proof in appendix.
> >
> > ## QUESTION.
> >
> > Good to have. Make the ML relevance more explicit.
> >
> > ## ANSWER.
> >
> > We hope the revamped Introduction which positions the work
> > at the cross-roads of clustering, cluster models, and center points, provides
> > a clear positioning in the realm of ML.
> >
> > Update(s): Section 1 i.e. Introduction.
> >
> > ## QUESTION.
> >
> > Good to have.  Expand the experimental comparison beyond BFGS/L-BFGS-B. Those baselines make sense for the solver story, but they do not say much about the value of the model itself. Comparisons to other robust center estimators would help.
> >
> > ## ANSWER.
> >
> > It is well admitted that solvers using sub-gradient descents are slower than L-BFGS-B,
> > especially in high dimension, which is why we focused on BFGS/L-BFGS-B.
> >
> > About the value of the model itself, see the next point.
> >
> > ## QUESTION.
> >
> > Good to have. Give a clearer practical interpretation of the parameter $\eta$.
> >
> > ## ANSWER.
> >
> > See answer above about the following statistics: *average outlier cost*, *outlier ratio* (wrt COM), and *Distance between points (wrt the projection median).
> >
> > Update(s): no update here since the statistics were already provided. See Section 5.2.
> >
> > ## QUESTION.
> >
> > Good to have. Tighten the writing and polish.
> >
> > ## ANSWER.
> >
> > The paper contained some typos we corrected and a few ambiguous phrasings, which we rewrote. We also clarified some proofs.
> >
> > ## QUESTION.
> >
> > Good to have. Clarify the role of the supplementary material.
> >
> > ## ANSWER.
> >
> > We opted for a presentation striking a balance between (i) the problem, (ii)
> > the mathematics properties of the functional, and (iii) experiments.
> > We believe that moving most of the technical content into the main text
> > would be detrimental for most readers -- those not versed into non smooth
> > optimization.

---

### Review · Reviewer_SFH1 · 2026-04-05

**Summary Of Contributions:**

This paper addresses a leftover of the spherical cluster (SC) model regarding the uniqueness of the center, and proposes new optimization algorithm to solve SC. The authors show that SC objective is strongly convex but non-smooth, and derive a geometric decomposition based on an arrangement of spheres. The algorithmic idea is tailored to the structure of the objective rather than borrowed off the shelf. Empirically, they compare the proposed solver to BFGS/L-BFGS-B on medium- and high-dimensional datasets, and argue that the SC center behaves like a parameterized high-dimensional median while often being much faster to optimize exactly than quasi-Newton heuristics.

**Audience:**

No

**Audience Explanation:**

I'm not sure about this option. Clustering is clearly an important problem for the ML community. However, as a paper centered on the optimization of an existing clustering pipeline, one needs especially strong evidence that this method is already valuable on its own. The SC cluster model explored in this work is not as widely adopted as famous ones like k-means and spectral clustering, so this paper doesn't fully clear that bar.

**Claims And Evidence:**

Yes

**Claims Explanation:**

The decomposition in Eq. (7) to (15) is useful and, at least at the level presented in the main paper, internally coherent. Theorem 1 claims strong convexity and uniqueness of the minimization problem for $0<\eta<1-\frac{1}{n}$. If correct, this is an important result because it turns what initially looks like a fairly awkward combinatorial non-smooth objective into a globally well-posed optimization problem. The fact that the paper does not merely propose a heuristic but analyzes existence and uniqueness is a real positive. The empirical comparison is focused on optimization behavior, which matches the paper’s main technical contribution.

**Requested Changes:**

The title "Modeling high dimensional point clouds..." and repeated uses of “cluster model” impose an initial impression as if the paper aims to derive a new model. However, the actual contribution is optimizing the SC objective. There is no evaluation of whether SC produces better cluster assignments (such topics are orthogonal to this work). Hence, the authors are suggested to refine the background and motivation in paper writing.

The complexity discussion is interesting but not very satisfying theoretically. The discussions in Section 4.3 and 5.5 may be true experimentally, but it leaves the reader without a meaningful complexity guarantee. As an optimization-centric paper, it could benefit more from theoretical complexity analytics.

---

> ### Author Response · Authors · 2026-04-28
> **Complements to our general answer.**
>
> ## QUESTION.
>
> The title "Modeling high dimensional point clouds..." and repeated uses of “cluster model” impose an initial impression as if the paper aims to derive a new model. However, the actual contribution is optimizing the SC objective. There is no evaluation of whether SC produces better cluster assignments (such topics are orthogonal to this work). Hence, the authors are suggested to refine the background and motivation in paper writing.
>
> ## ANSWER.
>
> We totally agree with this remark, which is also why the title of the paper does not contain *clustering*.
>
> We hope that the expanded Introduction, which discusses in turn *Clustering methods*,
> *Cluster models*, and *Geometric   centerpoints*, and provides a revised version of
> *Contributions* clarifies the situation.
>
>
> ## QUESTION.
>
> The complexity discussion is interesting but not very satisfying theoretically. The discussions in Section 4.3 and 5.5 may be true experimentally, but it leaves the reader without a meaningful complexity guarantee. As an optimization-centric paper, it could benefit more from theoretical complexity analytics.
>
> ## ANSWER.
>
> We added a paragraph discussing the merits and weaknesses of our method compared to other proofs.
>
> We also rewrote the main proof, and hope that this revised version provides insights
> on the ingredients making up the complexity.

---

### Review · Reviewer_DKYN · 2026-04-05

**Summary Of Contributions:**

This paper investigates the Spherical Cluster (SC) model, a parametric clustering approach designed to capture the geometric structure of high-dimensional data.

The SC model approximates a point set $P \subset \mathbb{R}^d$ using a sphere $S(c, r)$, where the radius $r$ is defined as $\eta$ times the standard deviation of the distances from the cluster center $c$ (for $\eta \in (0, 1)$). A penalty (power distance) is applied only to points residing outside the sphere. The center $c$ is defined as the point that minimizes this cost function. Notably, when $\eta = 0$, the model coincides with the standard center of mass used in K-Means.

The paper claims the following three primary contributions:

* Contribution 1: Mathematical Characterization. The authors characterize the optimization problem as a strictly convex yet non-smooth combinatorial optimization problem and prove the uniqueness and well-posedness of the solution.

* Contribution 2: Proposed Exact Solver. The authors propose SC-Exact-Solver, an exact iterative algorithm utilizing Clarke gradients and a semiflow on a stratified cell complex (an arrangement of hyperspheres). The solver consists of three main procedures: LineDescent, SphereDescent, and Teleportation.

* Contribution 3: Experimental Validation. Through experiments on various datasets ranging from $d=9$ to $d=10,000$, the authors confirm two key findings:
    * The proposed method is several orders of magnitude faster than BFGS-based approaches—specifically for small $\eta$ in low-to-medium dimensions, and across all $\eta$ values in high dimensions.
    * The center of the SC model behaves as a high-dimensional median parameterized by the hyperparameter $\eta$.

This work is positioned as the foundation for a forthcoming companion paper, which will introduce a mixture model of SCs embedded in affine subspaces.

**Additional Comments:**

### Critique on the role of Definition 1 and Definition 3:
The introduction of Definition 1 is conceptually meaningful in that it clarifies the transition from K-Means to the Spherical Cluster model. However, it does not directly contribute to the core findings of this paper. While it serves as a stepping stone toward Definition 3 (SESC), the analysis of SESC itself is explicitly stated to be outside the scope of this work. Consequently, both Definition 1 and Definition 3 function merely as "previews of future research," which makes the current structure of the paper feel somewhat redundant.

### Insufficient discussion on the dependency of the radius definition:
There is a lack of deep discussion regarding the dependency of the radius $r = \eta \sigma^{(c_\ell)}$ on the center $c_\ell$. Because the radius changes whenever the center is moved, the objective function takes on a highly complex structure. This dependency is the fundamental cause of the non-smoothness and the primary factor that makes the optimization problem challenging. The paper, however, does not sufficiently justify the necessity of this specific design choice.

**Audience:**

Yes

**Audience Explanation:**

High-dimensional data clustering is a classic and challenging problem in pattern recognition and machine learning, as it is directly impacted by the curse of dimensionality. Therefore, research that formulates this task as a (albeit non-smooth) convex programming problem and provides a practical algorithm is highly welcomed.
However, it is important to note that the study remains focused on evaluating the center point, rather than presenting a functional clustering method

**Claims And Evidence:**

Yes

**Claims Explanation:**

The proposed method is clearly explained and experimental results support the claims.

**Requested Changes:**

### Guidance on the selection of $\eta$ is required due to unclear criteria:
$\eta$ is a hyperparameter; however, the paper does not explicitly provide guidance on how to select it. In the experiments, $\eta$ is simply grid-searched across $\{0.1, 0.2, \dots, 0.9\}$, and the only suggestion for determining a practical value is to "examine the non-monotonicity of the outlier cost plot." A more formal or systematic guideline for its selection should be provided.

### Unclear justification for using power distance:
The statistical or geometric necessity of using power distance as the cost function is not sufficiently discussed. For instance, there is no comparison with the case using standard distance, such as $\max(0, \|x-c\| - r)$, making it unclear how the choice of power distance specifically impacts the results.

### Incomplete Definition 3:
Definition 3 is mathematically incomplete as it lacks the range of possible values, statistical/geometric interpretations, and selection criteria for $\mu$. These elements must be addressed and corrected to ensure the definition is rigorous.

### Relationship between Equation 7 and Equation 8:
Please clarify the relationship between $\tilde{f}_{\eta,x_i}$ in Equation 7 and $f_{\eta,x_i}$ in Equation 8.

### Concerns regarding experimental fairness (Asymmetry in comparison with BFGS):
There appears to be an asymmetry in the comparison between the proposed method and BFGS:
* SC-Exact-Solver: Utilizes a warm start (center of mass).
* SC-BFGS-Solver: The warm start setting is unclear, even though the choice of initial points significantly impacts the performance of BFGS.
Consequently, it is questionable whether this constitutes a fair comparison. Please provide a clarification regarding the experimental protocol to ensure fairness, and include supplementary data if necessary.

---

> ### Author Response · Authors · 2026-04-28
> **Complements to our general answer.**
>
> In addition to the comments provided in the 'General answer' section
> of the revision, the following comments are in order.
>
> ## QUESTION.
>
> Guidance on the selection of $\eta$ is required due to unclear criteria.
>
> ## ANSWER.
>
> The section 5.2 contained the definition of two statistics, the
> *average outlier cost* and the *outlier ratio* to guide the choice of $\eta$.
>
> The new Section 2.3 discusses the spherical cluster model in detail
> and mentions these statistics.  Comments on their typical behavior are
> provided at the end of section 5.3.
>
> To step back about this point, $\eta$ defines a one parameter family
> of models.  It could be of interest to try to derive general
> properties of the cost $F_{\eta}$ using hypothesis on the distribution of
> points processed -- in particular the noise component.
> This sounds like a hard problem.
>
> ## QUESTION.
>
> Unclear justification for using power distance.
>
> ## ANSWER.
>
> Explained in Section 2.3.
>
> ## QUESTION.
>
> Incomplete Definition 3
>
> ## ANSWER.
>
> This definition has been removed since the SESC model was not used.
>
>
> ## QUESTION.
>
> Relationship between Equation 7 and Equation 8
>
> ## ANSWER.
>
> Apologies: one latex macro had not been revamped. Fixed.
>
> ## QUESTION.
>
> Concerns regarding experimental fairness (Asymmetry in comparison with BFGS)
>
> ## ANSWER.
>
> The warm start is the same, namely the center of mass. This has been clarified when introducing BFGS.
>
> ## QUESTION.
>
> Critique on the role of Definition 1 and Definition 3
>
> ## ANSWER.
>
> We agree with this point and removed the definition of
> subspace embedded spherical clusters. We believe this helps focusing  the
> paper of the optimization problem of the spherical cluster model.
>
> ## QUESTION.
>
> Insufficient discussion on the dependency of the radius
>   definition.  Because the radius changes whenever the center is
>   moved, the objective function takes on a highly complex
>   structure. This dependency is the fundamental cause of the
>   non-smoothness and the primary factor that makes the optimization
>   problem challenging.
>
> ## ANSWER.
>
> This remark is touching the core of our paper. We believe the
> fundamental motivation for defining the spherical cluster model is
> rooted in statistical analysis: to identify an object capturing a
> global description of the point set.
>
> In our case, the center point  is defined to strike a balance between inliers (which pay zero) and outliers (which pay
> the power distance to the sphere).
>
> This has been clarified in the novel Section 2.3.  We also believe
> that the comparison between our outliers and those associated to a
> sphere centered at the center of mass warrant the choice of varying
> $c$.

---

### Decision · Action_Editor_ok5Q · 2026-05-24

**Recommendation:** Reject

**Audience:**

Yes

**Audience Explanation:**

High-dimensional point cloud modeling is an important topic, and some individuals in TMLR's audience will be interested in it.

**Claims And Evidence:**

No

**Claims Explanation:**

The proposed model-fitting problem is strongly convex but non-smooth, and the paper presents an efficient solver. However, most reviewers expressed negative opinions of this paper after the rebuttal.

The main claims that are not supported by accurate, convincing, and clear evidence include:

-  The main objective of this paper is one center for one cloud, but this paper is mainly presented as clustering.
-  The complexity discussion is built experimentally, but it could benefit more from theoretical complexity analytics.

---

> ### Author Response · Authors · 2026-06-12
> **Contesting the decision**
>
> Dear Editors in chief,
>
> I would like to contest the decision made by the Action Editor for two reasons.
>
> 1/ THE DECISION is NOT WARRANTED. We contest the arguments provided:
>
> 1a/ <<The main objective of this paper is one center for one cloud, but this paper is mainly presented as clustering.>>
>
> => The paper shows how to fit the so-called cluster model, a pre-requisite for clustering using this parametric cluster model.  Stating that our paper is presented as clustering is erroneous or fallacious. The claim  is an unwarranted statement copied from the review of the reviewer W2hj, who also refused to read our rebuttal, as shown by his reply
>
> <<Please address my comments and concerns here with pairs of question/concern and answer/edit. I cannot go back and forth between the revised paper and my comments here to find which edit is addressing which of my comments.>>
>
> 1b/ << The complexity discussion is built experimentally, but it could benefit more from theoretical complexity analytics.>>
>
> => Theorem 2 proves the algorithm convergence: its derivation is 10 page long NOT A SINGLE REMARK WAS MADE ABOUT IT.
> Stating that the complexity is experimental is nonsense. A valid argument could have been that the bound is not tight or is too local, but this is not the criticism being made. We are convinced our proof was not read and its subtlety was not appreciated.
>
>
> 2/ ACCEPTANCE CRITERIA: THERE IS NO EVIDENCE WE DO NOT MEET THEM.
>
> one reads at https://jmlr.org/tmlr/acceptance-criteria.html
>
> 2a/ Are the claims made in the submission supported by accurate, convincing and clear evidence?
>
> => We provide two theorems and a detailed experimental study. We got no technical complaint about these, just superficial/cosmetic ones.
>
> 2b/ Would some individuals in TMLR's audience be interested in the findings of this paper?
>
> => We address a statistical and non convex optimization problem in the context of clustering.  Specifically, one reads in the review of the reviewer XF2L: <<Therefore, this work is certainly of interest to at least some individuals in TMLR's audience. Specifically, the method's applicability to high dimensional data might be of interest.>>
>
>
> With best regards,
> Frederic Cazals.